# TAK1 mediates microenvironment-triggered autocrine signals and promotes triple-negative breast cancer lung metastasis

Oihana Iriondo[1,2], Yarong Liu[3], Grace Lee[1,2], Mostafa Elhodaky [1,2], Christian Jimenez[1,2], Lin Li[1,2], Julie Lang [2,4], Pin Wang[3] & Min Yu[1,2]

Triple-negative breast cancer (TNBC) is a highly metastatic subtype of breast cancer that has limited therapeutic options. Thus, developing novel treatments for metastatic TNBC is an urgent need. Here, we show that nanoparticle-mediated delivery of transforming growth factor-β1-activated kinase-1 (TAK1) inhibitor 5Z-7-Oxozeaenol can inhibit TNBC lung metastasis in most animals tested. P38 is a central signal downstream of TAK1 in TNBC cells in TAK1-mediated response to multiple cytokines. Following co-culturing with macrophages or fibroblasts, TNBC cells express interleukin-1 (IL1) or tumor necrosis factor-α (TNFα), respectively. Compared to TAK1 inhibition, suppressing IL1 signaling with recombinant IL1 receptor antagonist (IL1RA) is less efficient in reducing lung metastasis, possibly due to the additional TAK1 signals coming from distinct stromal cells. Together, these observations suggest that TAK1 may play a central role in promoting TNBC cell adaptation to the lung microenvironment by facilitating positive feedback signaling mediated by P38. Approaches targeting the key TAK1-P38 signal could offer a novel means for suppressing TNBC lung metastasis.

[1] Department of Stem Cell Biology and Regenerative Medicine, Keck School of Medicine, University of Southern California, Los Angeles, CA 90033, USA. [2] Norris Comprehensive Cancer Center, Keck School of Medicine, University of Southern California, Los Angeles, CA 90033, USA. [3] Department of Chemical Engineering and Materials Science, Viterbi School of Engineering, University of Southern California, Los Angeles, CA 90089, USA. [4] Department of Surgery, Keck School of Medicine, University of Southern California, Los Angeles, CA 90033, USA. These authors contributed equally: Grace Lee, Mostafa Elhodaky. Correspondence and requests for materials should be addressed to M.Y. (email: minyu@med.usc.edu)

Breast cancer is the most common cancer in women worldwide[1]. It is a heterogeneous disease with several subtypes corresponding to different treatment options and prognoses[2, 3]. Clinically, based on the status of estrogen receptor (ER), progesterone receptor (PR), and human epidermal growth factor receptor 2 (ERBB2/HER2)—patients are categorized into ER/PR+, HER2+, or triple-negative subtypes. Hormonal therapies and HER2 targeted therapies have led to dramatic improvement in the overall prognosis of ER/PR+ and Her2+ subtypes, respectively. However, the overall survival status of triple-negative breast cancer (TNBC) remains the worst among breast cancers and has remained stagnant over the past 20 years because of TNBC's aggressive nature and the lack of targeted therapies[4]. TNBC constitutes approximately 20% of breast cancer cases and has high rate of metastasis, particularly to the visceral organs, such as the lung, liver, and brain. Chemotherapy is still the standard treatment for both early and metastatic TNBC patients[4]. Thus, there is an urgent need to develop new therapeutic interventions for preventing and treating metastasis for the management of TNBC patients.

Transforming growth factor-β1 (TGF-β1)-activated kinase-1 (TAK1) is a mitogen-activated protein kinase kinase kinase (MAP3K7) that was discovered for its role in mediating TGFβ and bone morphogenetic protein (BMP) signaling[5]. Since then, it has been shown that TAK1 can be activated by a variety of cytokines and signals besides TGFβ, including interleukin-1 (IL1)[6], tumor necrosis factor-α (TNFα)[7], tumor necrosis factor apoptosis-inducing ligand[8], BMP[9], Wnt[10, 11], and lipopolysaccharide[12–14]. In response to these stimuli, TAK1 is involved in the inflammatory, innate immune response, and non-canonical Wnt pathway by activating a list of downstream signals, including P38[7, 13, 14], ERK[15], JNK[7, 13, 14], and NFκB[7, 12, 13].

Recently, TAK1 has been shown to play a role in tumorigenesis. We have previously demonstrated that targeting TAK1 can inhibit distant metastasis in a pancreatic cancer mouse model[16] and induce KRAS-dependent apoptosis in colorectal cancer[17]. Melisi et al.[18] have reported that inhibiting TAK1 can sensitize pancreatic cancer cells to the chemotherapeutic drug gemcitabine. In breast cancer, recent studies have shown that TAK1 plays important roles in mediating the effects of CCN6[19], TRAF4[20], UBC13[21], and mir-892b[22] on tumor progression and metastasis, and that targeting TAK1 enhances doxorubicin (DOXO)-mediated apoptosis and reduces invasive behavior[23, 24]. Analysis of TAK1 expression levels in breast tumors from the TCGA dataset using Oncomine showed a higher expression of TAK1 in TNBC compared to other subtypes (Supplementary Fig. 1a). This indicates that TAK1 is a potential candidate target for developing TNBC targeted therapy.

The tumor microenvironment plays a crucial role in cancer progression, with different signals from the microenvironment having tumor-promoting and tumor-suppressing effects[25]. During metastatic spread, tumor cells adapt to the microenvironment in the secondary organ and are likely to use local signals to activate specific pathways to promote tumor growth. Many of the TAK1-activating inflammatory cytokines are also present in the tumor microenvironment and involved in breast cancer progression[26]. For example, a higher level of IL1 has been correlated with more aggressive and high-grade breast cancers, and an elevated level of IL1's natural antagonist—recombinant IL1 receptor antagonist (IL1RA)—and a lower level of IL1 is a good prognostic indicator for breast cancer patients[27–31]. Interestingly, IL1α, IL1β, and IL1RA are expressed at a much higher level in TNBC cell lines compared to luminal cells[32], suggesting that the IL1 pathway could have an even more critical role in this subset of breast tumors. In other cancers—such as melanoma, lung, and colon carcinoma—IL1 has been shown to promote proliferation, inhibit

apoptosis, and induce angiogenesis[28, 29]. TGFβ and TNFα, which can also lead to TAK1 activation, are well recognized for their roles in tumorigenesis[26]. Thus, TAK1 may play a crucial role in mediating tumor cell interaction with the local microenvironment.

In previous studies, we demonstrated that 5Z-7-Oxozeaenol (OXO)[33], a potent ATP-competitive irreversible inhibitor for TAK1, had strong inhibitory effect against TAK1-mediated phenotypes in pancreatic cancer[16] and colorectal cancer[17]. However, the poor solubility of OXO limits its in vivo bioavailability and clinical utility in cancer therapy. Nanoparticle-mediated delivery of OXO could potentially solve this problem by prolonging its circulation half-life and improving its pharmacokinetics, thus resulting in enhanced therapeutic efficacy while reducing unwanted side effects.

Our previous application of cross-linked multilamellar liposomal vesicles (cMLVs) in a mouse melanoma model has demonstrated that these nanoparticles can achieve controlled delivery of cancer therapeutics with improved drug release kinetics and enhanced particle stability[34]. cMLVs exhibit an enhanced ability to accumulate in tumors, with fewer particles in the blood, heart, and spleen. In addition, cMLVs loaded with the anticancer drug DOXO have significantly improved therapeutic activity in inhibiting tumor growth[35]. Furthermore, cMLVs can encapsulate both hydrophobic and hydrophilic drugs[36]. Indeed, we have shown that nanoparticles loaded with DOXO (hydrophilic) and paclitaxel (hydrophobic) are highly efficient in inducing apoptosis in tumors and overcoming multidrug resistance[36].

In this study, we aimed to use cMLV nanoparticles to deliver the TAK1 inhibitor OXO and evaluate the role of TAK1 in TNBC lung metastasis, as well as decipher the upstream and downstream signaling that is crucial for TAK1-mediated lung metastasis in TNBC. We found that inhibition of TAK1 by different methods can reduce lung metastasis. Interactions with different components of the tumor microenvironment induce expression of distinct TAK1-activating cytokines in TNBC cells, which can lead to a positive feedback loop that amplifies TAK1 signaling, promoting metastatic growth.

## Results

**OXO nanoparticles can suppress lung metastasis.** To study the effect of TAK1 inhibition on established primary tumors and metastatic growth, we used cMLVs loaded with the TAK1 inhibitor OXO. We also used nanoparticles loaded with DOXO, a chemotherapy drug widely used in breast cancer treatment, as a control for drug efficiency. Both OXO and DOXO could efficiently be released from cMLVs in vitro (Supplementary Fig. 1b and Fig. 1a), leading to dose-dependent decreases in the viability of MDA-MB-231 cells—comparable to those achieved when cells were treated with free OXO or DOXO (Supplementary Fig. 1c, d).

To study the effect of TAK1 inhibition on established primary breast tumors, MDA-MB-231 cells tagged with green fluorescent protein (GFP) and luciferase (MDA-MB-231-GFP/luc) were injected into the mammary fat pads of NSG (NOD.Cg-Prkdc^scid Il2rg^tm1Wjl/SzJ) mice. Fourteen days later, mice were randomly assigned to four groups, each of which received cMLVs containing empty vehicle (EV), OXO, DOXO, or the combination of both drugs (OXO + DOXO). After 30 days of treatment, as expected, mice treated with DOXO had smaller tumors compared to the control mice. In contrast, tumors from OXO-treated mice were similar to those found in control mice (Fig. 1b, c), showing that TAK1 inhibition does not impair the growth of established primary tumors.

We next evaluated whether cMLV-encapsulated OXO has any effect on lung metastatic lesions. MDA-MB-231-GFP/luc cells

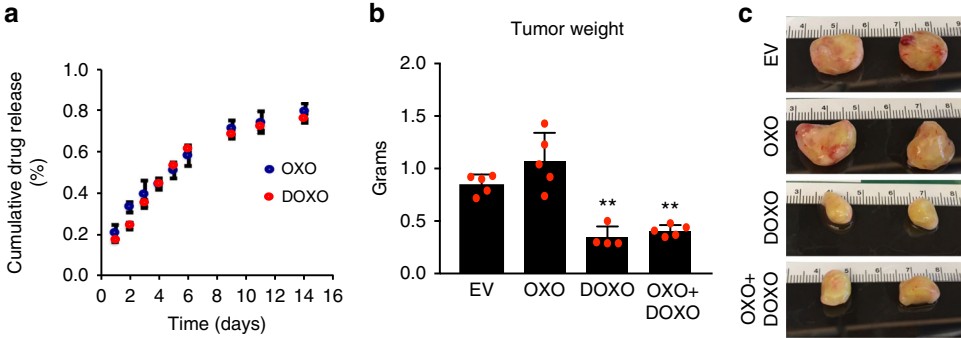

**Fig. 1** Treatment with 5Z-7-Oxozeaenol (OXO) does not reduce growth of established primary tumors. **a** In vitro release kinetics of OXO and doxorubicin (DOXO) from dual drug-loaded cross-linked multilamellar vesicles (cMLVs). Means ± SD of three measurements are shown. **b**, **c** Average tumor weight (**b**) and representative images (**c**) of primary tumors dissected from mice treated with nanoparticles containing empty vehicle (EV), OXO, DOXO, or the combination of both drugs for 30 days. Treatment with nanoparticles was initiated 2 weeks after orthotopically inoculating mice with two million MDA-MB-231 cells. Data are shown as means ± SEM of five mice. Statistical analysis: unpaired two-tailed Student's $t$ test. **$p \leq 0.01$ compared to empty vehicle (EV) control

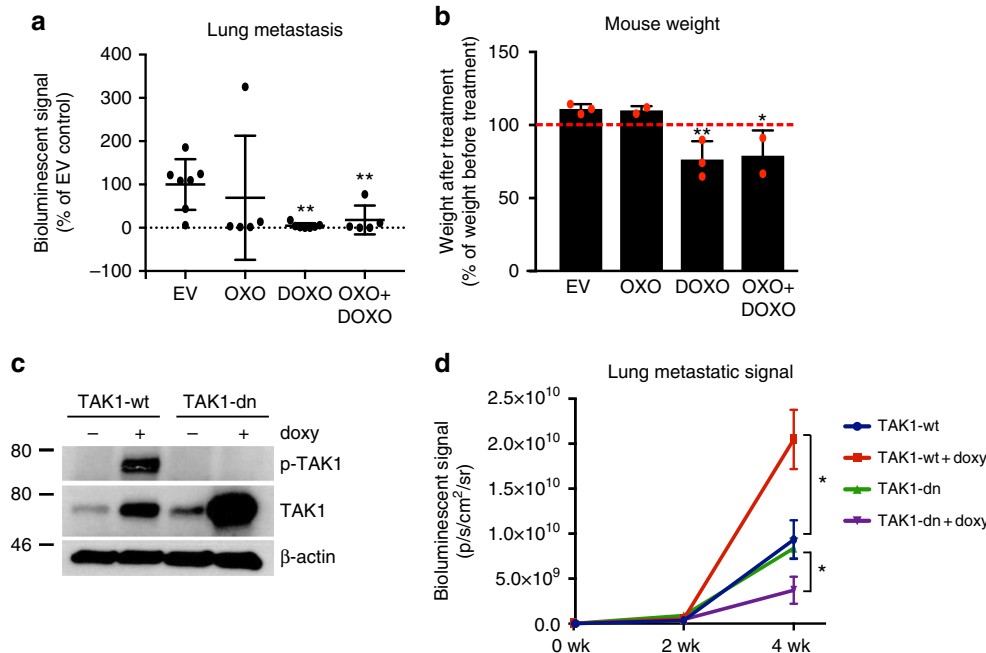

**Fig. 2** TAK1 inhibition reduces metastatic growth in the lung. **a** Luminescent signal of lungs dissected from mice treated with nanoparticles containing empty vehicle (EV), 5Z-7-Oxozeaenol (OXO), doxorubicin (DOXO), or a combination of both drugs. Primary tumors were resected 2 weeks after orthotopically inoculating mice with two million luciferase-overexpressing MDA-MB-231 cells, and treatment with drug-containing nanoparticles was initiated 1 day after primary tumor removal. Averages ± SD of seven mice (EV and DOXO) or five mice (OXO and OXO + DOXO) are shown. **b** Changes in body weight of mice treated with nanoparticles containing EV, OXO, DOXO, or a combination of both drugs. For each mouse, change in weight was calculated by comparing the weight at the end of the experiment with that on the day that treatment with nanoparticles started. Averages ± SD of three (EV and DOXO) or two (OXO and OXO + DOXO) mice are shown. **c** TAK1 and P-TAK1 expression in MDA-MB-231 cells that overexpress wild-type TAK1 (TAK1-wt) and a dominant-negative form of TAK1 (TAK1-dn) in a doxycycline-inducible manner. A representative example of three experiments is shown. **d** Lung bioluminescent signal in mice injected with MDA-MB-231 cells expressing doxycycline-inducible TAK1-wt and TAK1-dn with or without treatment of doxycycline (doxy). Data are represented as averages ± SEM of five mice. Statistical analyses: unpaired one-tailed Student's $t$ test with Welch's correction for unequal SDs (**a**) and unpaired two-tailed Student's $t$ test (**b** and **d**). In **a** and **b**, treatment groups are compared to control group (EV). In **d**, signals at 4 weeks of mice injected with each cell line (TAK1-wt or TAK1-dn) and treated with doxycycline are compared to signals of mice injected with the same cell line that did not receive doxycycline. *$p \leq 0.05$; **$p \leq 0.01$

were orthotopically implanted in NSG mice and allowed to grow for 2 weeks, after which primary tumors were resected. Starting 1 day after primary tumor removal, mice were treated with EV, OXO, DOXO, or OXO and DOXO containing cMLVs for 30 days. To eliminate the influence of residual primary tumors, the effect on lung metastasis was evaluated in mice with clean primary tumor removal (see Methods). The metastatic signal in

the mouse lungs was drastically reduced in most of the mice treated with OXO (Fig. 2a). Mice treated with DOXO also showed reduced bioluminescence signals in the lungs; however, unlike OXO, treatment with DOXO showed significant toxicity for the mice, demonstrated by decreased body weight and impaired kidney and liver functions, while mice treated with OXO showed little to no sign of toxicity (Fig. 2a, b and Supplementary Fig. 2a).

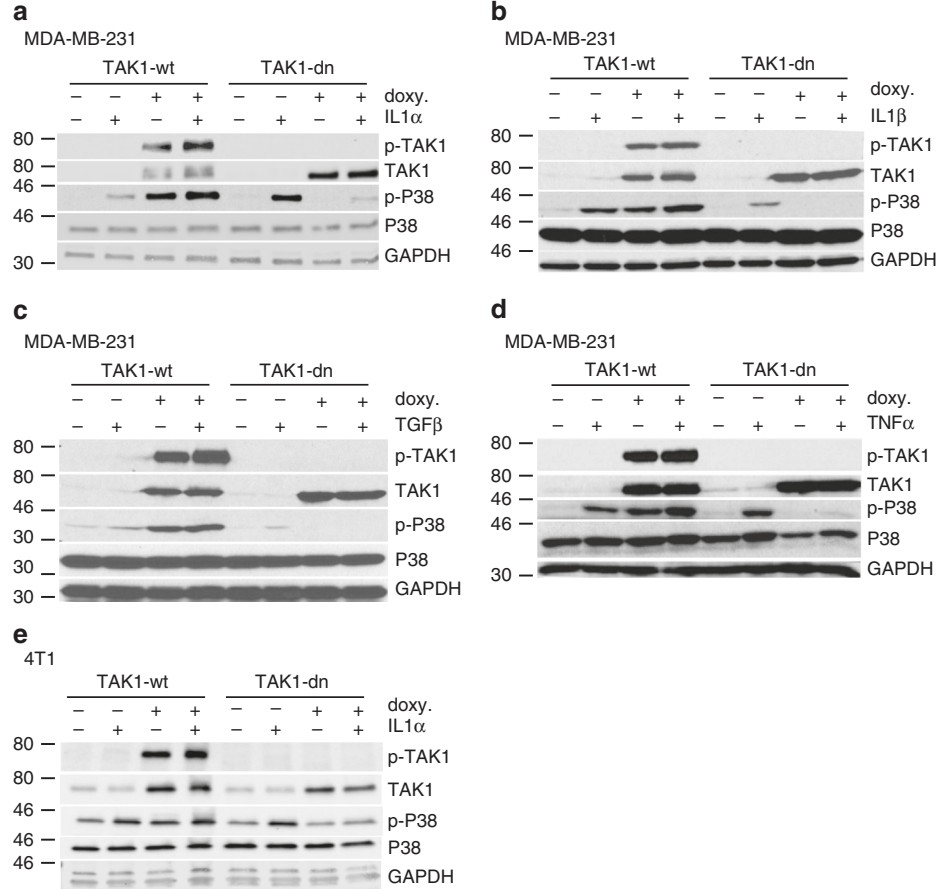

**Fig. 3** TAK1 regulates P38 phosphorylation in triple-negative breast cancer cells. **a–d** P38 phosphorylation in MDA-MB-231 cells overexpressing doxycycline-inducible wild-type TAK1 (TAK1-wt) or dominant-negative TAK1 (dn-TAK1) and treated with IL1α (**a**), IL1β (**b**), TGFβ (**c**), and TNFα (**d**). **e** P38 phosphorylation in 4T1 cells overexpressing doxycycline-inducible TAK1-wt or TAK1-dn and treated with IL1α. In all cases, cells were plated in suspension using serum-free media. Representative examples of three experiments are shown

In order to confirm that the decrease in lung metastasis observed in mice treated with OXO was due to its ability to inhibit TAK1, we used MDA-MB-231-GFP/luc cells overexpressing wild-type TAK1 (TAK1-wt) or a dominant-negative form of TAK1 (TAK1-dn) in a doxycycline-inducible manner (Fig. 2c). TAK1-dn has a point mutation (K63W) in the ATP-binding domain that abrogates its kinase activity. Mutant TAK1 can still bind to TAB1 and TAB2/3, competing with endogenous TAK1 and thus acting as a dominant-negative form[7]. TAK1-wt and TAK1-dn MDA-MB-231-GFP/luc cells were cultured in the presence or absence of doxycycline for 2 days to induce the overexpression of either form of TAK1. Cells were then injected into the tail veins of NSG mice. Mice injected with the doxycycline-treated cells received doxycycline in their drinking water, and the bioluminescent signal was measured every 2 weeks. As shown in Fig. 2d, overexpression of wild-type TAK1 increased lung metastasis, while overexpression of dominant-negative TAK1 reduced it (Fig. 2d and Supplementary Fig. 2b).

**TAK1/P38 mediate responses to cytokines**. Depending on the cellular and microenvironmental context, TAK1 can lead to the activation of several pathways, including P38 MAPK, ERK1/2 MAPK, and NFκB[37]. To test which signaling pathways are influenced by TAK1 in TNBC cells, we treated MDA-MB-231 cells overexpressing inducible wild-type or dominant-negative TAK1 with several cytokines known to activate TAK1[5–7] that could have relevant roles in lung metastasis[26]. In order to avoid

possible interference by cytokines present in the fetal bovine serum (FBS) of our regular culture media, the experiment was first performed in cells cultured in suspension conditions in FBS-free media. Cells cultured in the presence or absence of doxycycline for 2 days were treated with IL1α (Fig. 3a), IL1β (Fig. 3b), TGFβ (Fig. 3c), or TNFα (Fig. 3d). In all four cases, the increase in P38 phosphorylation in response to the cytokines was abrogated by the overexpression of the dominant-negative TAK1. In addition, overexpression of wild-type TAK1 increased P38 phosphorylation levels, both in the presence and absence of cytokine stimulation (Fig. 3a-d ). In contrast, ERK1/2 and P65 phosphorylation did not change significantly after inhibiting TAK1 activity by the overexpression of TAK1-dn (Supplementary Fig. 3a). Blocking TAK1 activity also prevented cytokine-induced P38 phosphorylation in 4T1 and HS578T cells (Fig. 3e and Supplementary Fig. 3b), showing that TAK1 activity is required for P38 phosphorylation in several TNBC cell lines. Similar effects in P38 phosphorylation were found when the experiment was performed in regular cell culture conditions. However, in this case, P65 phosphorylation in response to cytokines was partially prevented by overexpression of dominant-negative TAK1, although the effect was not as pronounced as for P38 phosphorylation (Supplementary Fig. 3c, d).

**Macrophages induce IL1 expression in tumor cells**. We analyzed the cytokine sources in the normal NSG mouse lung and found that they were more prominently expressed by stromal

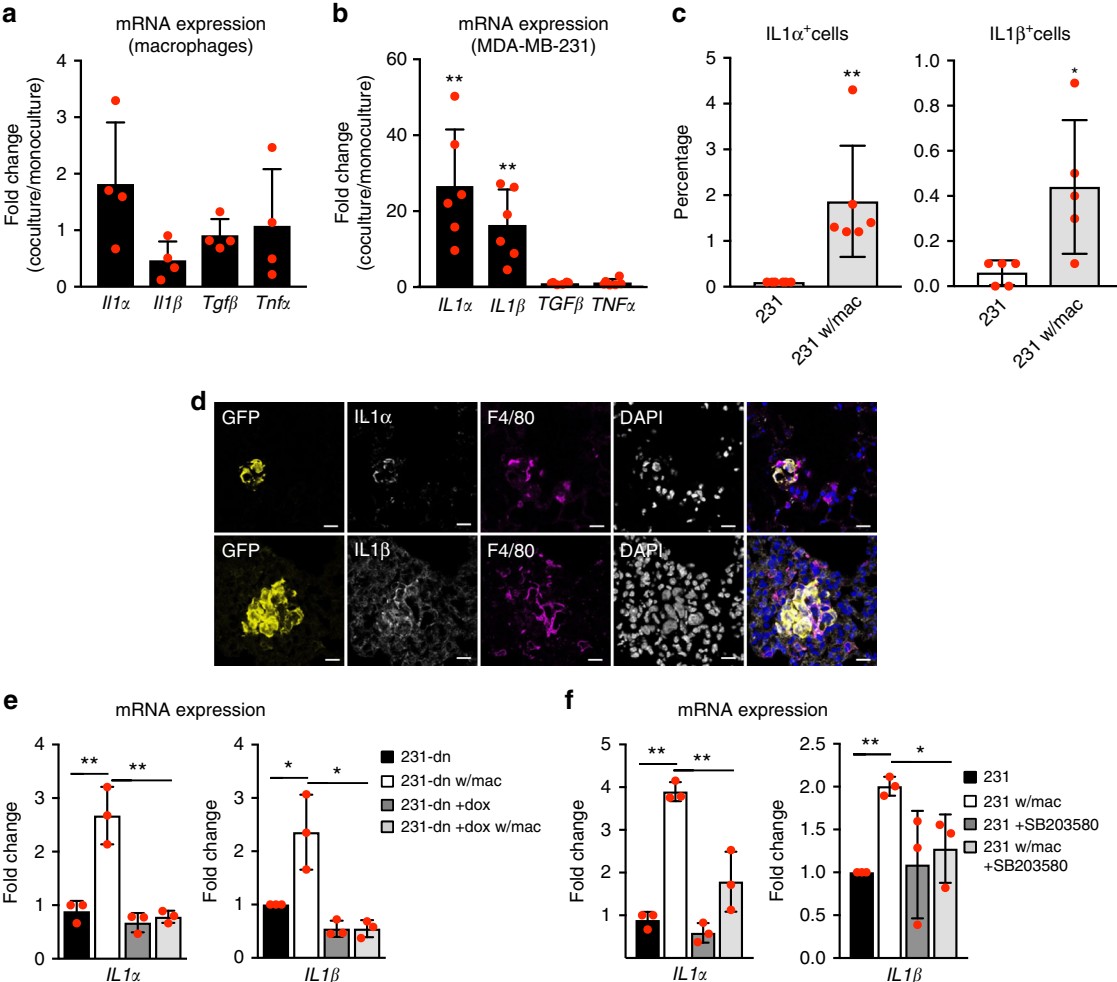

**Fig. 4** Co-culture with macrophages increases IL1α and IL1β expression in MDA-MB-231 cells in a TAK1-dependent manner. **a** qPCR analysis of *Il1α*, *Il1β*, *Tgfβ*, and *Tnfα* expression in macrophages cultured in the presence or absence of MDA-MB-231 cells for 5 days. **b** qPCR analysis of *IL1α*, *IL1β*, *TGFβ*, and *TNFα* expression in MDA-MB-231 cells cultured in the presence or absence of macrophages for 5 days. **c** Percentage of IL1α-positive (left) and IL1β-positive (right) cells in GFP-tagged MDA-MB-231 monocultures and GFP-tagged MDA-MB-231 cells cultured with macrophages. **d** Immunofluorescent staining of lungs dissected from mice injected with GFP and luciferase-tagged MDA-MB-231 cells. Representative examples of images taken from three immunostainings of two independent experiments are shown. GFP (yellow), macrophage marker F4/80 (magenta), IL1α or IL1β (white), DAPI (gray in single channel, blue in merge image) are shown. Scale bar = 20 μm. **e** qPCR analysis of *IL1α* and *IL1β* expression in MDA-MB-231 cells stably expressing doxycycline-inducible dominant-negative TAK1 (TAK1-dn), grown in co-culture with macrophages for 3 days in the presence or absence of doxycycline (doxy). **f** qPCR analysis of *IL1α* and *IL1β* expression in MDA-MB-231 cells cultured in the presence or absence of macrophages and treated or not with 1 μM SB203580. In **a**, **b**, **e**, and **f**, GFP-tagged MDA-MB-231 cells were used, and macrophages and MDA-MB-231 cells were separated by FACS for RNA extraction. Graphs show means ± SD of three (**e**, **f**), four (**a**), five (**c**, right panel) and six (**b**, **c**, left panel) independent experiments. Statistical significance analyzed by unpaired two-tailed Student's *t* test *$p \leq 0.05$; **$p \leq 0.01$

cells (Supplementary Fig. 4a). Given the known role of macrophages in tumor tissues[25], we hypothesized that macrophages recruited to lung metastatic lesions could express TAK1-activating cytokines upon interacting with cancer cells. To test this in vitro, GFP-tagged MDA-MB-231 cells were cultured with RAW264.7 murine macrophages for 5 days, after which cancer cells and macrophages were separated by fluorescence-activated cell sorting (FACS) based on GFP expression. Quantitative real-time PCR (qPCR) analysis showed that mRNA expression levels of *Il1α*, *Il1β*, *Tgfβ*, and *Tnfα* in macrophages did not significantly change when these cells were co-cultured with MDA-MB-231 cells (Fig. 4a). In contrast, MDA-MB-231 cells cultured with macrophages upregulated the expression of *IL1α* and *IL1β* at the mRNA level, whereas *TGFβ* and *TNFα* expression levels remained unchanged (Fig. 4b). Intracellular flow cytometry showed that, on average, 1.86% and 0.44% of MDA-MB-231 cells

co-cultured with macrophages expressed IL1α and IL1β at the protein level, respectively (Fig. 4c and Supplementary Fig. 4b), suggesting that only a small subset of MDA-MB-231 cells respond to signals from macrophages by increasing the expression of these cytokines. Immunofluorescence analysis of lungs of NSG mice injected with MDA-MB-231 cells showed that cancer cells surrounded by macrophages express IL1α and IL1β as early as 5 days post injection (Fig. 4d and Supplementary Fig. 4c). The increase in *IL1α* and *IL1β* in response to the co-culture with macrophages depends on TAK1 and P38 activity, since it is prevented when TAK1 activity is blocked by overexpressing dominant-negative TAK1 (Fig. 4e) or by treating cells with the P38 inhibitor SB203580 (Fig. 4f and Supplementary Fig. 4d). In contrast, despite the fact that macrophages used for the co-culture experiment secrete high levels of TNFα (Supplementary Fig. 4e), blocking the NFκB pathway using PS1145 does not influence the

upregulation of *IL1α* and *IL1β* by the co-culture condition (Supplementary Fig. 4f, g). Cancer cells treated with recombinant IL1α or IL1β upregulate the expression of those same cytokines (Supplementary Fig. 4h), suggesting that secretion of IL1α or IL1β in a few cancer cells could lead to the upregulation of both cytokines in neighboring cancer cells. These experiments suggest that TAK1-P38-mediated upregulation of IL1α and IL1β as a consequence of interactions with macrophages establishes a positive feedback loop involving TAK1 and P38 that would lead to enhanced activity of the pathway, favoring metastatic growth. Corroborating our results, P38 has been shown to be critical for TNBC lung metastasis[19, 21].

**Targeting IL1 is less sufficient in suppressing metastasis.** Several types of solid tumors—including breast, colon, lung, head, and neck cancers and melanoma—overexpress IL1β[29]. In breast cancer, IL1α and IL1β expression has been correlated with the lack of ER expression, high tumor grade, and poor differentiation[27, 30, 31]. Based on these findings and our results showing that tumor cells upregulate IL1α and IL1β when in contact with macrophages, we hypothesized that increased IL1 expression could promote lung metastasis by activating the TAK1-P38 pathway. To test this, we used the interleukin 1 receptor antagonist (IL1RA) anakinra, which is Food and Drug Administration-approved for the treatment of rheumatoid arthritis, and shows no toxicity in mice (Supplementary Fig. 5a, b). We first confirmed the effect of IL1RA by western blot, showing that MDA-MB-231 cells treated with IL1RA are unable to increase P38 phosphorylation in response to treatment with IL1α (Fig. 5a). Next, MDA-MB-231-GFP/luc cells overexpressing TAK1-dn were injected into the tail veins of NSG mice after being cultured with or without doxycycline for 2 days, and the effect of IL1 receptor inhibition on cancer metastasis was evaluated. Control mice and mice receiving doxycycline in drinking water were intraperitoneally injected with phosphate-buffered saline (PBS) or 1.5 mg/kg anakinra daily, starting 2 days before the injection of cancer cells, and lung metastatic burden was evaluated 4 weeks later by bioluminescence imaging. Consistent with the previous results shown in Fig. 2d, inactivation of TAK1 activity by overexpressing dominant-negative TAK1 significantly reduced metastasis formation (Fig. 5b). However, although mice treated with anakinra showed a tendency towards decreased metastatic burden, the difference did not reach statistical significance ($p = 0.08$) (Fig. 5b). We next sought to determine if blocking TAK1 and/or IL1 pathways could influence metastatic growth in immune competent mice, using the murine breast cancer cell line 4T1 in syngeneic BALB/c mice. 4T1 cells treated with IL1RA also failed to induce P38 phosphorylation in response to IL1α treatment (Fig. 5c). Upon injection of luciferase-tagged 4T1-TAK1-dn cells, mice that received doxycycline treatment developed significantly fewer metastases than control mice. Anakinra treatment also decreased lung metastatic signals with a difference approaching statistical significance ($p = 0.06$). Mice in which both TAK1 and IL1 signaling were blocked by the administration of both doxycycline and anakinra showed significantly lower metastatic burden than mice that received a single treatment (Fig. 5d). The fact that TAK1 inhibition significantly reduced metastatic growth in both mouse models, while IL1 inhibition had a smaller effect could mean that cancer cells in the lung microenvironment are provided with other signals that can activate the TAK1-P38 pathway. Indeed, we found that MDA-MB-231 cells cultured in the presence of NIH/3T3 fibroblasts upregulate mRNA expression levels of TNFα (Fig. 5e), which can increase P38 phosphorylation in a TAK1-dependent manner (Fig. 3d). In addition, TNFα was detected in lung

micrometastasis of mice injected with MDA-MB-231-GFP/luc cells (Fig. 5f and Supplementary Fig. 5c).

**Discussion**

In this study, we have shown that cMLV nanoparticle-mediated delivery of the TAK1 inhibitor OXO could potentially suppress the development of lung metastasis formed by TNBC cells without detectable toxicity. P38 is the predominant signaling induced by TAK1 in response to a list of cytokines. TAK1-P38 signaling enables an autocrine positive feedback loop to induce TAK1-activating cytokine expression in tumor cells. The specific cytokines induced in the tumor cells depend upon the stromal cells with which tumor cells interact. These data suggest that TAK1-P38 is a central signaling pathway facilitating TNBC cell adaptation in the lung, providing a plausible rationale for developing therapeutic interventions involving this pathway for managing TNBC lung metastasis.

The cMLVs loaded with OXO showed an inhibitory effect on lung metastasis development in mouse models in which primary tumors had been cleanly removed. With the exception of one mouse which showed high metastatic signal, all other animals treated with the TAK1 inhibitor had decreased metastatic signals that were similar to those in mice treated with DOXO. The effect of OXO is validated by overexpression of wild-type or dominant-negative TAK1. However, the dramatic difference is the lack of toxicity to the animals treated by OXO in contrast to DOXO. For patients with TNBC, treatment options remain limited to systemic chemotherapies, and there are few novel targeted therapies currently in clinical trials in TNBC, despite a compelling need for new agents. In addition, patients suffer from toxicity associated with systemic chemotherapies. Considering the toxicity of chemotherapy in patients and the efficiency of cMLV-mediated delivery of OXO in reducing lung metastasis in mice, we believe that nanoparticle-mediated delivery of the TAK1 inhibitor is an approach that could potentially improve treatment of TNBC lung metastasis.

P38 is central to TAK1 function in TNBC cells in response to various cytokines. This is different from the findings in immune cells during the inflammatory response, where, in response to different inflammatory cytokines, TAK1 activates an array of downstream pathways, including NFκB, P38, ERK, JNK, etc[7]. In TNBC cells, we found that P38 is the predominant signal downstream of TAK1 and is critical for autocrine IL1 expression. Importantly, others have shown that the inhibitor against P38, SB203580, suppressed TNBC lung metastasis in a mouse model[21].

We did not see an inhibitory effect on established primary tumors, suggesting that TAK1 signaling is not essential for maintaining tumor growth once primary tumors have been established. This finding is similar to several previous reports showing no effect on primary tumor growth by targeting upstream or downstream TAK1 pathways[21, 24]. This could be due to the distinct microenvironments of the primary tumor and the lung, resulting in different dependence on TAK1 function. Interestingly, a study has shown that depleting macrophages using CSF-1R inhibitors in primary breast tumors has no effect on primary tumor growth, unless used in combination with another treatment[38]. In contrast, the same approach showed a robust suppression of glioma[39]. These different responses in breast and brain cancers suggest the possible divergent influences of these cells in distinct organs.

Our most intriguing finding is that when TNBC cells are in contact with macrophages or fibroblasts, there is microenvironment-triggered increase in IL1 or TNFα levels, respectively, which can lead to autocrine positive feedback loops in the cancer cells. TAK1-P38 signaling is crucial for IL1

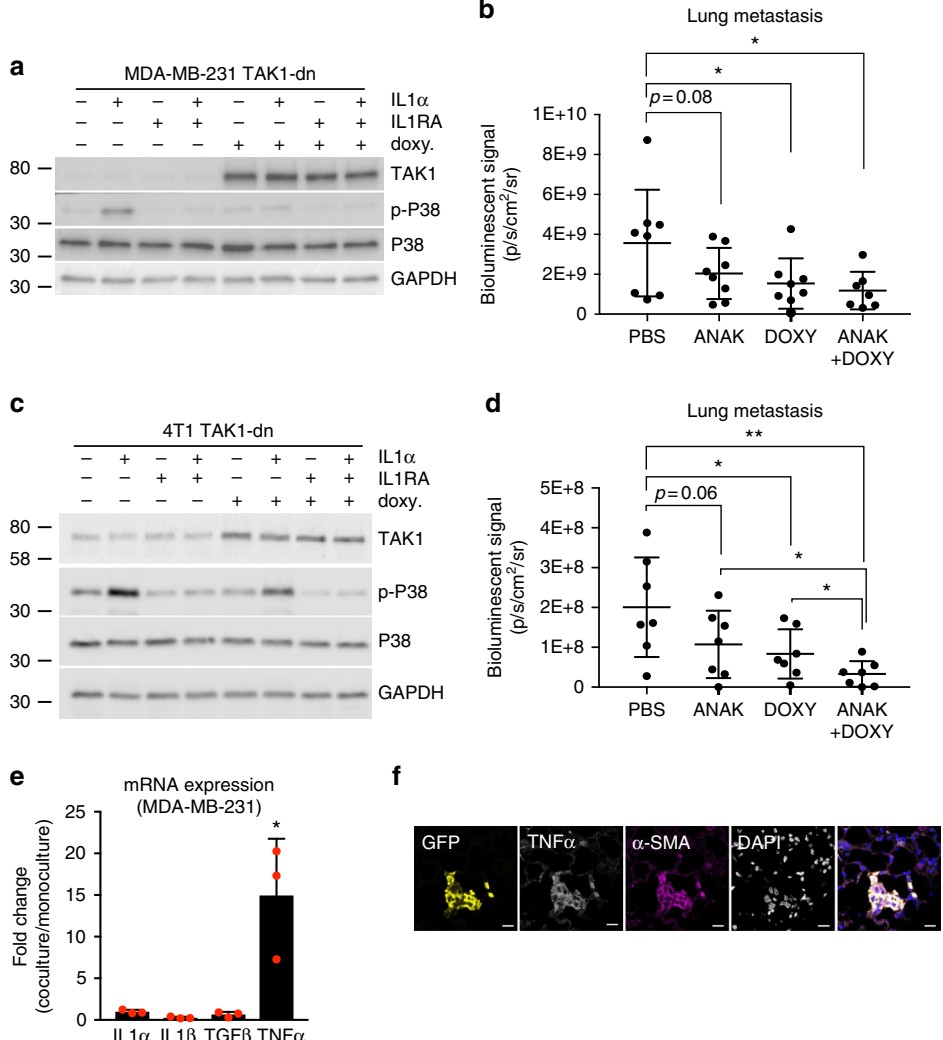

**Fig. 5** Treatment with anakinra could reduce lung metastasis, but less so than TAK1 inhibition. **a**, **c** Changes in P38 phosphorylation levels in response to IL1α (10 ng/ml) in MDA-MB-231 cells (**a**) and 4T1 cells (**c**) overexpressing doxycycline-inducible dominant-negative TAK1 (TAK1-dn) with or without treatment of doxycycline and recombinant IL1RA (100 ng/ml). Representative examples of three experiments are shown. **b**, **d** Lung metastatic burden determined by measuring the bioluminescent signal in mice injected with MDA-MB-231 (**b**) and 4T1 (**d**) cells overexpressing doxycycline-inducible TAK1-dn and with or without treatment with doxycycline and/or anakinra. Bioluminescence was measured 4 and 3 weeks after tail vein injection of MDA-MB-231 and 4T1 cells, respectively. Scatter dot plots show means ± SD of signals obtained from eight (**b**) or seven (**d**) mice per group. **e** qPCR analysis of *IL1α*, *IL1β*, *TGFβ*, and *TNFα* expression in MDA-MB-231 cells cultured in the presence or absence of GFP-expressing fibroblasts for 5 days. RNA was extracted from cells separated by FACS based on GFP expression. Means ± SD of three independent experiments are shown. **f** Immunofluorescent staining of lungs dissected from mice injected with GFP and luciferase-tagged MDA-MB-231 cells. Representative examples of images taken from three immunostainings of two independent experiments are shown. GFP (yellow), myofibroblast marker α-SMA—also expressed in MDA-MB-231 cells (magenta), TNFα (white), DAPI (gray in single channel, blue in merge image), are shown. Scale bar = 20 µm. Statistical analyses: unpaired one-tailed Student's *t* test (**b**, **d**), unpaired two-tailed Student's *t* test (**e**). *$p \leq 0.05$; **$p \leq 0.01$

autocrine signaling. Once IL1 pathway is activated in some cells, the signal can be amplified and spread to neighboring cells. To our knowledge, this is the first report showing a TAK1-P38-mediated positive feedback loop in TNBC cells. This finding underscores the complex interactions between tumor cells and stromal cells in the tumor microenvironment, revealing multiple means to active the TAK1 signal. Thus, suppressing either one of the upstream cytokines is not sufficient to completely block TAK1-mediated downstream signaling and metastatic growth. The results of the experiments we performed using the IL1 receptor inhibitor anakinra also point in this direction. In both mouse models tested, the group of mice treated with anakinra showed a tendency towards decreased metastatic burden, but the effect was not as pronounced as when TAK1 was inhibited, either

with nanoparticle-encapsulated OXO (if we exclude the outlier sample) or by overexpressing the dominant-negative TAK1. We do not know which molecule in stromal cells triggers the positive feedback loop in tumor cells. Even though there is a clear expression of TNFα by macrophages, NFκB inhibitors did not have any effect in blocking the positive feedback loop of IL1.

A growing body of data has demonstrated the contribution of the tumor microenvironment to tumor growth and progression[25]. It is a general consensus that metastasis formation is regulated to a large extent by interactions of cancer cells with the microenvironment[25, 40]. Tumor cells deposited in a new organ, such as lung, must be able to adapt to the new environment and overcome the suppressive signals to support their growth. In this complex network, TAK1 seems to play a crucial role. TAK1 is a

well-known factor that plays a central role in immune cells during inflammation. It is a key kinase linking cellular responses to exogenous stimuli. Thus, it is not surprising to see that tumor cells also utilize TAK1 in adapting to their new microenvironments. In human patients, it is likely that the microenvironment is even more complex than immunodeficient xenograft mouse models and includes additional immune cells that reside or are recruited to lung metastatic sites. These immune cells could augment the crucial role of TAK1 in mediating this metastatic adaptation. It is tempting to speculate that TAK1 is an essential cell adaptation kinase, responding to local cytokines and forming positive feedback loops to support its signaling, at least in TNBC metastasizing to the lung. Thus, targeting TAK1 or downstream P38 could be effective in inhibiting lung metastasis. Whether TAK1 plays a similar role in metastases to other organs needs further investigation. Of note, a study showed that TAK1 provided a suppressive signal for breast cancer to bone metastasis downstream of HGFK1[41], contrary to most reports in lung metastasis, suggesting that the role of TAK1 could be particularly important in lung metastases. In addition, another unanswered question is whether the role of TAK1 in TNBC is heterogeneous among different molecular subclasses of TNBC, due to their distinct molecular properties[42].

## Methods

**Cell culture and reagents**. MDA-MB-231, 4T1, HS578T, HEK293T, RAW264.7, and NIH/3T3 cells were purchased from the American Type Culture Collection. MDA-MB-231, HS578T, HEK293T, RAW264.7, and NIH/3T3 cells were grown in Dulbecco's modified Eagle's media supplemented with 10% FBS, while 4T1 cells were grown in RPMI media containing 10% FBS. None of the cell lines is listed as commonly misidentified cell lines by ICLAC. All cell lines were routinely tested for mycoplasma contamination.

To study the involvement of TAK1 in activating different signaling pathways, cells cultured in the presence or absence of 1 μg/ml doxycycline (Clontech) for 48 h were treated with recombinant IL1α (10 ng/ml, 10 min), IL1β (10 ng/ml, 10 min), TGFβ (2 ng/ml, 5 min), or TNFα (20 ng/ml, 10 min). All cytokines were purchased from R&D Systems. For in vitro experiments using recombinant human IL1RA (Peprotech), cells were treated with 100 ng/ml IL1RA for 30 min, followed by treatment with IL1α (10 ng/ml) for an additional 10 min. For experiments done in suspension conditions, cells were plated in poly-hema-coated dishes in RPMI media supplemented with B27 (Invitrogen), 20 ng/ml epidermal growth factor (Peprotech), and 20 ng/ml basic fibroblast growth factor (Peprotech).

Co-culture experiments were performed by plating MDA-MB-231 cells stably expressing GFP and luciferase and RAW264.7 macrophages or non-tagged MDA-MB-231 cells and GFP-expressing 3T3 fibroblasts by themselves or in combination. After 3 or 5 days (see figure legends), cells were trypsinized and separated based on GFP expression by FACS, and RNA was extracted from the sorted cells. In some cases, cells were treated with 1 μg/ml doxycycline, 2 μM PS1145 (R&D System, 975108), or 1 μM SB203580 (Abcam, ab146589) from the beginning of the experiment.

**Generation of stable cell lines**. The doxycycline-inducible expression constructs containing wild-type or K63W mutant TAK1 were generated by subcloning each open reading frame into the pInducer 20 lentiviral vector[43] using the Gateway cloning system (Thermo Fisher Scientific). For virus packaging, HEK293T cells were co-transfected with the lentiviral expression plasmid, the packaging construct pCMVdeltaR8.91, and the vesicular stomatitis virus glycoprotein G (VSV-G) envelope construct using TransIT-LT1 (Mirus). Viral supernatants collected 48 and 72 h post transfection were combined, filtered through 0.45 μM polyethersulfone filters, and stored in aliquots at −80 °C. The lentiviral plasmid expressing GFP-luciferase (GFP-luc) was obtained from C. Ponzetto (University of Torino, Italy) and co-transfected with third-generation lentivirus packaging plasmids (PMDL, REV, and VSV-G) using Lipofectamine 2000. Viral supernatants collected 48 h after transfection were filtered, concentrated using Lenti-X-concentrator (Takara), and stored at −80 °C. Transductions were carried out in the presence of 8 μg/ml polybrene (Sigma). Cells infected with pInducer 20 viruses were selected with 1 mg/ml geneticin (Invitrogen), while cells expressing the GFP-luc construct were selected by FACS.

**Preparation of cMLVs**. All lipids were obtained from NOF Corporation (Japan): 1,2-dioleoyl-*sn*-glycero-3-phosphocholine (DOPC), 1,2-dioleoyl-*sn*-glycero-3-phospho-(10-rac-glycerol) (DOPG), and 1,2-dioleoyl-*sn*-glycero-3-phosphoethanolamine-*N*-[4-(*p*-maleimidophenyl) butyramide (maleimide-headgroup lipid, MPB-PE). DOXO and OXO were purchased from Tocris Bioscience.

Liposomes were prepared based on the conventional dehydration–rehydration method. All lipids were combined in chloroform at a molar lipid ratio of DOPC −DOPG−MPB = 4:1:5, and the chloroform in the lipid mixture was evaporated under argon gas. The lipid mixture was further dried under a vacuum overnight to form dried thin lipid films. In order to prepare cMLVs (DOXO + OXO), OXO was mixed with the lipid mixture in organic solvent before the formation of the dried thin lipid films. The resultant dried film was hydrated in 10 mM Bis-Tris propane at pH 7.0 with DOXO by vigorous vortexing every 10 min for 1 h. Four cycles of 15 s sonication were then applied (Misonix Microson XL2000, Farmingdale, NY, USA) on ice at 1 min intervals for each cycle. To induce divalent-triggered vesicle fusion, MgCl₂ was added at a final concentration of 10 mM. The resulting multilamellar vesicles were further cross-linked by the addition of dithiothreitol (Sigma-Aldrich), at a final concentration of 1.5 mM for 1 h at 37 °C. The resulting vesicles were collected by centrifugation at $14,000 \times g$ for 4 min and then washed twice with PBS. For pegylation of cMLVs, the particles were incubated with 1 μmol of 2 kDa PEG-SH (Laysan Bio Inc. Arab, AL, USA) for 1 h at 37 °C. The particles were then centrifuged and washed twice with PBS. The final products were stored in PBS at 4 °C.

**In vitro drug encapsulation and release**. To obtain the release behavior of DOXO and OXO from cMLVs, the releasing media was removed from cMLVs incubated in 10% FBS-containing media at 37 °C and replaced with fresh media every other day. The removed media were quantified for DOXO fluorescence (by spectrofluorometer) and OXO fluoresence (by high-performance liquid chromatography) after different retention times (flow rate 1 ml/min).

**In vitro cytotoxicity assay**. MDA-MB-231 cells were seeded in 96-well plates at a concentration of $5 \times 10^3$ cells per well and incubated at 37 °C for 6 h. Cells were then exposed to a series of concentrations of single drug-loaded cMLVs for 48 h. Fifty microliters of XTT labeling mixture (Roche Applied Science) was then added. After incubation at 37 °C for 4 h, the absorbance of the solution was measured at 570 nm using a microplate reader (Molecular Devices) to determine the optical density value. Cell viability was calculated by subtracting absorbance values obtained from media-only wells from drug-treated wells and then normalizing to the control cells without drugs. Data are given as mean ± SD of three independent measurements.

**Assessment of in vivo toxicity**. Mouse blood was collected through cardiac puncture and centrifuged at $14,000 \times g$ for 3 min. Serum was transferred to new tubes and immediately frozen. Blood urea nitrogen, alanine aminotransferase, and aspartate aminotransferase were measured using colorimetric kits according to the manufacturer's instructions.

**Mouse xenograft tumor assays**. The animal protocol was approved by the Institutional Animal Care and Use Committee of the University of Southern California. For mouse experiments using nanoparticles, 6-week old female NSG mice (Jackson Laboratory) were anesthetized with isofluorane and two million MDA-MB-231-GFP/luc cells in 100 μl of 1:1 PBS and Matrigel (BD) were injected into the fourth mammary fat pad. To evaluate the effect of OXO and DOXO on the growth of established primary tumors, mice were separated into four groups 2 weeks after tumor cell injection, and each group was treated with empty vesicles or nanoparticles containing OXO (2 mg/kg), DOXO (2 mg/kg), or the combination of both drugs by tail vein injection every 3 days for 4 weeks, after which tumor weight was measured. To evaluate the effect of the drugs upon metastasis formation, primary tumors were resected 2 weeks after orthotopic injection of MDA-MB-231-GFP/luc cells. Mice were randomized according to residual primary tumor signal and mouse weight, assigned to one of four groups, and treated with empty vesicles or nanoparticles containing OXO, DOXO, or the combination of both every 3 days, starting the day after tumor resection. Metastatic load was evaluated by ex vivo imaging of lungs. Mice were intraperitoneally injected with 150 μl of D-Luciferin substrate at 30 mg/ml (Sid Labs), and sacrificed after 5 min, and their lungs were dissected and imaged using IVIS Lumina II (Perkin Elmer). Only mice for which complete tumor resection was achieved were considered for the analysis. This was evaluated by looking at images taken shortly after primary tumor removal.

For tail vein injection, 200,000 MDA-MB-231-GFP/luc or 100,000 4T1-luc cells overexpressing doxycycline-inducible wild-type or mutant forms of TAK1 were injected into the lateral tail vein of 7-week-old or 8-week- old female NSG or BALB/c mice, respectively. Signals from luciferase-tagged cells were monitored weekly by in vivo imaging. Doxycycline was supplied in drinking water (with 1% sucrose). The IL1RA anakinra (Kineret, Sobi) was injected intraperitoneally on a daily basis at 1.5 mg/kg, starting 2 days before tumor cell injection. For experiments with doxycycline and anakinra treatments, mice were randomized according to their weight.

Investigators were not blinded to group allocation during the experiment and when assessing the outcome of the experiment. No statistics were applied to determine sample size.

**Western blotting**. For western blot analysis, cells were lysed in Laemmli buffer (50 mM Tris, pH 6.8, 1.25% sodium dodecyl sulfate, 15% glycerol). Cell extracts were heated at 95 °C for 15 min for a complete lysis and denaturalization. After protein quantification with Lowry protein assay (Bio-Rad), 5% (v/v) β-mercaptoethanol (Sigma), and bromophenol-blue (Sigma) were added to each lysate. Proteins were separated on 4–15% polyacrylamide gradient gels (Bio-Rad) and transferred to polyvinylidene difluoride (PVDF) membrane (Millipore). Primary and secondary antibodies used are listed in Supplementary Table 1 and Supplementary Table 2, respectively. Uncropped versions of western blots from main figures are shown in Supplementary Fig. 6.

**Fluorescent immunohistochemistry**. Lungs were fixed in 4% paraformaldehyde for 4.5 h at 4 °C, washed three times with cold PBS, incubated in 30% sucrose at 4 °C overnight, and embedded in optimal cutting temperature (OCT). Frozen OCT blocks were prepared using dry ice/ethanol baths and stored at −80 °C. Seven micromolar sections were used for immunostaining with antibodies listed in Supplementary Table 1. Secondary antibodies used are shown in Supplementary Table 2. Images were taken using a Leica SP8-8X confocal microscope.

**Intracellular flow cytometry**. MDA-MB-231-GFP/luc cells were plated by themselves or in co-culture with RAW264.7 macrophages. After 5 days, cells were labeled with Zombie UV fixable viability dye (BioLegend), fixed and permeabilized using Fixation/Permeabilization Solution Kit (BD Biosciences), and stained with phycoerythrin-conjugated antibodies for IL1α or IL1β. Cells were analyzed on a BD LSR II (Becton Dickinson). Information about the antibodies used is available in Supplementary Table 1.

**Quantitative real-time PCR**. Total RNA was extracted using Quick-RNA MicroPrep (Zymo) and reverse transcribed with iScript Reverse Transcription Supermix (Bio-Rad), according to the manufacturer's instructions. qPCR was performed with a CFX96 Real-Time PCR Detection System (Bio-Rad), using iTaq Universal SYBR Green Supermix. Fold changes were calculated by the $\Delta\Delta$CT method, using 36B4 as the control. The primers used are shown in Supplementary Table 3.

**Statistical analysis**. Statistical analysis was performed using the GraphPad Prism. Data from at least three independent experiments are expressed as means ± SD unless stated otherwise. Normal distribution was assumed and Student's $t$ test was applied for two-component comparisons, using Welch correction when groups with different variances were compared. One-tailed statistical analyses were performed in mouse experiments addressing the efficacy of a treatment in reducing metastatic burden. Two-tailed analyses were used for other experiments. Confidence level of 0.95% was used for all statistical analyses ($\alpha$-value = 0.05). No statistics was applied to determine sample size.

**Data availability**. All data generated or analyzed during this study are included in this published article (and its Supplementary Information files).

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

## Acknowledgements

We thank Gohar Saribekyan and Maria Donaldson for technical assistance and Cristy Lytal for editing the manuscript. This work was supported by grants from National Cancer Institute (NCI) K22 CA175228-01A1 (M.Y.), National Cancer Institute (NCI) DP2CA206653 (M.Y.), USC MHI for Engineering Medicine for Cancer (M.Y., P.W., J.L.), and National Cancer Institute (NCI) P30CA014089 (core facilities).

## Author contributions

O.I. and M.Y. conceived, designed, and conducted the study, analyzed data, and prepared the manuscript. J.L. and P.W. were involved in the conceptual design. Y.L. synthesized and characterized nanoparticles and performed toxicity studies. L.L and Y.L. participated in the design and performance of the in vivo experiments with nanoparticles. O.I. performed in vitro and in vivo experiments. G.L., M.E., and C.J. assisted with in vivo experiments and performed in vitro experiments.

## Additional information

**Competing interests:** The authors declare no competing interests.

