## [Peer Review File · Nature Communications]

Reviewers' comments:

Reviewer #1 (Remarks to the Author):

This manuscript describes an effort to develop a clinical inhibitor of TAK1. In a previous work, the investigators discovered 5Z-7-Oxozeaenol as an inhibitor of TAK1. In this publication, it is formulated as a nanoparticle to improve pharmacologic properties. In xenograft models, this compound doesn't affect growth of established tumors, but it often decreases pulmonary metastasis. This leads to the current investigation of mechanism of TAK1 signaling. This occurs via P38 phosphorylation. Also there is an autocrine IL1 expression that regulates TAK1 signals. IL1 α / β expression is induced by macrophages. Inhibiting IL1 signals with anakinra partially decreases this signaling and lowers lung metastasis modestly.

Overall, this is a well-written manuscript and most of the conclusions are well supported. It is an important addition to the literature as it illustrates the role of TAK1 and IL autocrine loops in promoting pulmonary metastasis in mouse models. Additionally, the study does include 2-3 cell lines. While this does not sample the high variability of TNBC, it does indicate some generalizability of the findings. The primary concern is that the abstract and discussion (esp first sentence) concludes that 5Z-7-Oxozeaenol can 'efficiently suppress the development of lung metastasis.' This describes Figure 2A where there is a large error in the measure due to one outlying point. Statistically the conclusion isn't valid, although it is acknowledged that there perhaps the outlier is an anomaly. The conclusion is probably correct, but given the data, it should be stated in more circumspect fashion.

Major:

1. Suggest that the conclusions are moderated about lung metastasis, based on Figure 2A, as mentioned above.
2. The immunofluorescence experiments (Fig. 4C and 5E) are uncontrolled. There should be a negative control--e.g. cultured cells without macrophages imaged on the same scale. Ideally there should be quantitation of the fluorescence. All IF images should have scale bars-- (I'm unable to determine if I am seeing a small number or many cells in a clump because the scale is unclear). DAPI would be easier to see if lighter or grayscale for the single channel.
3. Excluding the IF data, the increase in IL1 α / β is only shown on the RNA level. Immunoblot or ELISA is probably the best way to quantitate protein, but if that is not done, it could be more clear that (line 334-5) the microenvironment-triggered increase in IL1 and TNF α is shown for mRNA, not protein.
4. The conclusion at the end of the discussion (penultimate sentence) is that further PKs are needed before moving into clinical trials. I'm curious what clinical situation the authors imagine for clinical trials? Since they do not shrink metastasis, would this be used for adjuvant therapy? That would be a particularly challenging situation in which to test a new therapy.

Minor:

- ER/PR are described as cell-surface receptors in the abstract and the text whereas they are actually nuclear receptors.
- gemcitabine is misspelled on line 85

Reviewer #2 (Remarks to the Author):

In this manuscript, the authors demonstrate utility of targeting TAK1 in TNBC using nanoparticle mediated delivery of 5Z-7-Oxozeaenol as a means to inhibit metastasis. Their data very nicely demonstrate that this method of delivery of the inhibitor delivers the compound to tumors, and while it does not inhibit primary tumor growth (a bit surprising given previous data on inhibition of

TAK1 in breast cancer), it dramatically inhibits metastasis with limited toxicity. They further elucidate the mechanism by which TAK1 mediates its effects on metastasis, focusing on Map Kinase P38, and the response of the TAK1-p38 pathway to cytokines induced by macrophages or fibroblasts. Data very nicely support their hypothesis, and experiments are rigorously performed. While the novelty of this particular finding is not that high (given that TAK1 has already been implicated in TNBC metastasis (Huang et al., *Oncotarget* 2015) and 5Z-7-Oxozeaenol has also already been shown to have anti-tumor effects in TNBC (Zhang et al., *Chem Biol Drug Des* 2017; Huang et al., *Oncotarget* 2015; Acuna et al., *Anticancer Res* 2010), the authors do significantly increase our understanding of how the pathway is actually active and promotes metastasis (on a molecular level) and also find novel ways to deliver the compound to show convincing in vivo data (also a new approach). Thus, overall this paper does significantly contribute to moving this into a more pre-clinical realm and increasing our understanding of how the pathway acts. Minor comments include the following:

1. The TCGA data discussed to show that TAK1 expression is higher in TNBC than in other subtypes should be shown (not just stated as "data not shown").
2. On line 164, the comment "mice were treated the same way as for primary tumors" is confusing- it's not clear what is meant by that.
3. How was "clean" primary tumor removal assessed?
4. Figures could use better labeling throughout so that one does not need to refer to the text or figure legend to understand exactly what is being shown in the figure.
5. The experiment with RAW264.7 cells to show macrophage involvement would be better if performed with primary macrophages.
6. In Fig. 5B, is the combination treatment better than either treatment alone?
7. In Fig. 5C, why does the TAK1-dn not inhibit P-p38? This seems counter to the argument throughout the whole paper.

Reviewers' comments:

Reviewer #1 (Remarks to the Author):

This manuscript describes an effort to develop a clinical inhibitor of TAK1. In a previous work, the investigators discovered 5Z-7-Oxozeaenol as an inhibitor of TAK1. In this publication, it is formulated as a nanoparticle to improve pharmacologic properties. In xenograft models, this compound doesn't affect growth of established tumors, but it often decreases pulmonary metastasis. This leads to the current investigation of mechanism of TAK1 signaling. This occurs via P38 phosphorylation. Also, there is an autocrine IL1 expression that regulates TAK1 signals. IL1 α / β expression is induced by macrophages. Inhibiting IL1 signals with anakinra partially decreases this signaling and lowers lung metastasis modestly.

Overall, this is a well-written manuscript and most of the conclusions are well supported. It is an important addition to the literature as it illustrates the role of TAK1 and IL1 autocrine loops in promoting pulmonary metastasis in mouse models. Additionally, the study does include 2-3 cell lines. While this does not sample the high variability of TNBC, it does indicate some generalizability of the findings. The primary concern is that the abstract and discussion (esp first sentence) concludes that 5Z-7-Oxozeaenol can 'efficiently suppress the development of lung metastasis.' This describes Figure 2A where there is a large error in the measure due to one outlying point. Statistically the conclusion isn't valid, although it is acknowledged that there perhaps the outlier is an anomaly. The conclusion is probably correct, but given the data, it should be stated in more circumspect fashion.

We thank the reviewer for the positive feedback and suggestion on modifying the conclusion regarding Fig. 2A. We agree that the conclusion should closely reflect the data and we have therefore modified the text (pages 2 and 14).

Major:

1. Suggest that the conclusions are moderated about lung metastasis, based on Figure 2A, as mentioned above.

Indeed, we are aware that not all animals from the Oxozeaenol treatment had reduced metastatic signal, and we have attempted to make that clear by showing all individual data points in the graph. However, we agree that the conclusions both in the abstract and the discussion are too strong considering the data shown. Therefore, we introduced modifications in both sections (pages 2 and 14). We believe that the mouse with high metastatic signal is an outlier. We further discuss this when answering to the third question of the second reviewer.

2. The immunofluorescence experiments (Fig. 4C and 5E) are uncontrolled. There should be a negative control--e.g. cultured cells without macrophages imaged on the same scale. Ideally there should be quantitation of the fluorescence. All IF images should have scale bars-- (I'm unable to determine if I am seeing a small number or

many cells in a clump because the scale is unclear). DAPI would be easier to see if lighter or grayscale for the single channel.

We agree that the immunofluorescence experiments we show are uncontrolled. However, we think that comparing them to a staining done with cultured cells will not solve the problem. The immunofluorescence protocols for staining tissue sections and cells on coverslips are very different (e.g. lungs are fixed for 4.5 hours with 4% PFA, while cells are fixed for 10 minutes), which would influence the intensity of the staining. We thought that comparing cells with or without contact with macrophages in the same tissue section would be more appropriate. Thus, we tried to quantify the percentage of cells with high IL1 α or IL1 β signal in tumor cells that were in contact or close to macrophages and those that were not. However, we could hardly see any tumor cells that did not have any macrophages close to them, making this quantification impossible. Although not ideal, we have provided the negative control with the secondary antibody staining in the Supplementary Fig. 4C to show the negative signal with the same staining procedure and imaging criteria. In addition, scale bars were added and the DAPI images were changed to make them easier to see.

3. Excluding the IF data, the increase in IL1alpha/beta is only shown on the RNA level. Immunoblot or ELISA is probably the best way to quantitate protein, but if that is not done, it could be more clear that (line 334-5) the microenvironment-triggered increase in IL1 and TNFalpha is shown for mRNA, not protein.

We agree that this is a valid point. To address this question, we first performed western blot using both the cell extracts and the supernatants of MDA-MB-231 monocultures and co-cultures with RAW264.7 cells, using antibodies that we had validated by confirming that they were able to detect recombinant human IL1 α and IL1 β and TNF α . However, we were unable to detect any of the cytokines (both in their mature and precursor forms). We reasoned that the increase in mRNA levels of the cytokines that we see could be due to a few cells in the co-cultures expressing them at the time point we tested, which could make it difficult to detect in a bulk lysate analysis. To address that possibility, we performed intracellular flow cytometry and found that, indeed, an average of 1.86% and 0.44% MDA-MB-231 cells start expressing IL1 α and IL1 β respectively when cultured in the presence of macrophages (Fig. 4C and Supplementary Fig.S4B). Although the percentage of cells that express IL1 α and IL1 β at the protein level is very low at this time point, we show that treatment of MDA-MB-231 cells recombinant IL1 α or IL1 β leads to increased mRNA levels of both cytokines (Supplementary Fig. S4G), suggesting that when a few cells start expressing those cytokines, the effect could be amplified by a positive feedback loop. These results are discussed in the revised manuscript on pages 10 and 11 (results) and page 16 (discussion).

4. The conclusion at the end of the discussion (penultimate sentence) is that further PKs are needed before moving into clinical trials. I'm curious what clinical situation the authors imagine for clinical trials? Since they do not shrink metastasis, would this be

used for adjuvant therapy? That would be a particularly challenging situation in which to test a new therapy.

We agree with the reviewer that moving the TAK1-mediated therapy at current stage would be challenging. Indeed, if clinical trials for this potential therapy were to be designed, we would suggest using the anti-TAK1 treatment as adjuvant therapy in patients that underwent surgery. Our data showed that TAK1 inhibition did not shrink the primary tumors. Our current study also did not directly test whether suppression of TAK1 will lead to the shrinkage of already established macrometastases. Instead, our data suggest the importance of TAK1-mediated signals and self-amplifying autocrine loops in the establishment of lung metastasis. Although we believe that these data provided important mechanistic understanding of this pathway, we agree with the reviewer that its translational potential at this point is still very complicated. Thus, we have now removed this particular sentence in the discussion.'

Minor:

-ER/PR are described as cell-surface receptors in the abstract and the text whereas they are actually nuclear receptors.

We apologize for the wrong usage of the word. The text has been corrected.

-gemcitabine is misspelled on line 85

We thank the reviewer for pointing the typo. The text has been corrected.

Reviewer #2 (Remarks to the Author):

In this manuscript, the authors demonstrate utility of targeting TAK1 in TNBC using nanoparticle mediated delivery of 5Z-7-Oxozeaenol as a means to inhibit metastasis. Their data very nicely demonstrate that this method of delivery of the inhibitor delivers the compound to tumors, and while it does not inhibit primary tumor growth (a bit surprising given previous data on inhibition of TAK1 in breast cancer), it dramatically inhibits metastasis with limited toxicity. They further elucidate the mechanism by which TAK1 mediates its effects on metastasis, focusing on Map Kinase P38, and the response of the TAK1-p38 pathway to cytokines induced by macrophages or fibroblasts. Data very nicely support their hypothesis, and experiments are rigorously performed. While the novelty of this particular finding is not that high (given that TAK1 has already been implicated in TNBC metastasis (Huang et al., Oncotarget 2015) and 5Z-7-Oxozeaenol has also already been shown to have anti-tumor effects in TNBC (Zhang et al., Chem Biol Drug Des 2017; Huang et al., Oncotarget 2015; Acuna et al., Anticancer Res 2010), the authors do significantly increase our understanding of how the pathway is actually active and promotes metastasis (on a molecular level) and also find novel ways to deliver the compound to show convincing in vivo data (also a new approach). Thus, overall this paper does significantly contribute to moving this into a more pre-clinical realm and increasing our understanding of how the pathway acts. Minor comments include the following:

We thank the reviewer for his/her positive evaluation and suggestions.

1. The TCGA data discussed to show that TAK1 expression is higher in TNBC than in other subtypes should be shown (not just stated as "data not shown").

The oncomine analysis of TCGA data has been added as Supplementary Fig.1A.

2. On line 164, the comment "mice were treated the same way as for primary tumors" is confusing- it's not clear what is meant by that.

We have now specified how the treatment was done in order to make the procedure we followed clear (page 8 in Results section, page 23 in Methods section).

3. How was "clean" primary tumor removal assessed?

The graph shown in Fig. 2A combines results obtained in two experiments that were conducted in the exact same way. Two million cells were injected in the MFP and tumors were allowed to grow for 2 weeks, after which primary tumors were resected. After primary tumor removal, mice were imaged to evaluate the presence of any residual signal. This information was used to assign mice to different treatment groups, making sure that those mice with different levels of residual signal were evenly distributed among groups. The first experiment started with 5 mice in each group. From those, the number of mice that lacked any residual signal based on bioluminescent images after primary tumor removal were 4 (EV), 3 (OXO), 4 (DOXO) and 3 (OXO+DOXO). After analyzing this experiment, we realized that the presence of residual primary tumor seemed to interfere with the treatment, and we decided to repeat the experiment in order to get more mice with clean primary tumor removal. Because mice treated with doxorubicin seemed to have lost weight by the end of the experiment we decided to monitor treatment toxicity in the next experiment. For the second experiment, the initial number of mice was 7 for EV and DOXO groups and 8 for OXO and OXO+DOXO groups, while only 3 (EV), 2 (OXO), 3 (DOXO) and 2 (OXO+DOXO) had complete primary tumor removal. The graph in Fig. 2A shows the result obtained with all mice in which primary tumor was completely removed (EV = 7 mice, OXO = 5 mice, DOXO = 7 mice and OXO+DOXO = 5 mice). Also note that in the second experiment, 4 mice (1 from OXO group, 1 from DOXO group and two from OXO+DOXO group) died before the end of the experiment, due to imaging accident in OXO group and potential toxicity from doxorubicin in other groups. A brief description of the criteria used for sample exclusion has been added to the text (page 24).

In the process of answering to this question, we tried a different way of analyzing all the raw data. Instead of using the presence or absence of residual signal to filter the data, we used Graphpad software to identify outliers, using the ROUT method. Applying a stringent Q value (Q=0.2%) 0/12 EV, 2/12 OXO, 1/10 DOXO, and 3/11 OXO+DOXO outliers are detected. Looking at the residual signals of those 6 outliers detected, we found that 3 of them had the first, second and third highest residual signal in experiment 1, and 2 of them had the highest and the third highest residual signal of experiment 2. We believe this observation confirms that poor primary tumor resection is linked to the presence of those outliers. Removing the outliers detected with the above-described

method, we would get the following graph (means and SD are shown) that we believe supports our conclusion.

4. Figures could use better labeling throughout so that one does not need to refer to the text or figure legend to understand exactly what is being shown in the figure.

We changed the labeling in several figures to make them more self-explanatory.

5. The experiment with RAW264.7 cells to show macrophage involvement would be better if performed with primary macrophages.

We have tried using M-CSF to differentiate primary monocytes into macrophages to do the co-culture with tumor cells, but we did not see the result we expected. The reason for this may be that we did not use the macrophage that is able to activate TAK1 in the cancer cells. It is well known that the interactions of monocytes and macrophages with tumor cells *in vivo* are very complex. Functionally different cells from the monocytic lineage have different effects in tumor progression. For example, Pollard and colleagues showed that Ly6C⁺ inflammatory monocytes, which are preferentially recruited to pulmonary metastases, play a seminal role in the establishment of these metastases (Qian *et al.*, *Nature* 2011). In contrast, Ly6C⁻ patrolling monocytes participate in cancer surveillance by preventing metastasis to the lung (Hanna *et al.*, *Science* 2015). Regarding macrophages, alternatively activated or “M2” macrophages have been associated with tumor progression (Quail and Joyce, *Nature Medicine* 2013). However, a recent publication showed that tumor associated macrophages did not express M2 markers (Franklin *et al.*, *Science* 2014). M-CSF stimulation leads to an anti-inflammatory M2-like phenotype (Italiani and Boraschi, *Front Immunol* 2014) and it is possible that macrophages with this phenotype do not express the combination of cytokines that will result in TAK1 activation in cancer cells.

Furthermore, macrophage biology in the context of metastatic tumor microenvironment is extremely complicated. As above mentioned, even the widely used definition of “M1” and “M2” subtypes of macrophages and their role in tumor progression is being debated. It is also poorly understood when, where and how during differentiation of bone marrow derived monocytes toward macrophages do the monocytes/macrophages interact with tumor cells at distant metastatic sites. Thus, although we think it will be interesting to figure out what subpopulation of macrophages can lead to TAK1 activation in tumor cells, this may need a whole new study to do it properly. We believe that, ideally, this study would have to be done completely *in vivo*, to make sure that the

phenotypic and functional characteristics of different macrophage populations are not modified by *in vitro* culture conditions.

Despite the lack of this particular information, we now show with new data that the effect of TAK1-mediated upregulation of IL1 seems to self-amplify in tumor cells. Even if there is only a small subpopulation of tumor cells that upregulate IL1 after contacting macrophages, the effect can be spread into other tumor cells. With the new data we added, we think the importance of TAK1-mediated autocrine signaling and its complex nature in the context of the interactions with the tumor microenvironment will add on our knowledge of lung metastasis and raise interest in the research communities studying the tumor microenvironment, TAK1 signaling and metastasis.

6. In Fig. 5B, is the combination treatment better than either treatment alone?

The differences between the combination treatment and doxycycline and anakinra treatments are not statistically significant. However, difference between anakinra and combination treatment is closer to statistical significance (p value=0.08 for anakinra vs. doxycycline+anakinra, p value=0.27 for doxycycline vs. doxycycline+anakinra). Although we are aware that lack of statistical significance should prevent us from making a strong conclusion from these data, it could indicate that IL1 inhibition on top of TAK1 inhibition does not result in an additive effect, because inhibiting TAK1 already blocks downstream effectors of the IL1 pathway. Conversely, the efficacy of the combination treatment could be better than that of IL1 inhibition by itself, because IL1 inhibition does not influence the activation of TAK1 by other cytokines such as TNF α .

7. In Fig. 5C, why does the TAK1-dn not inhibit P-p38? This seems counter to the argument throughout the whole paper.

The western blot in the previous Fig.5C was performed with cells that had been in culture for a late passage after introduction of TAK1-dn. We repeated the same experiment with newly thawed cells that were used for the *in vivo* experiment and we think the result is better represented in the new Fig. 5C. However, the overexpression of TAK1-dn in 4T1 cells is much lower compared to that in MDA-MB-231 cells. Thus, the effect of TAK1-dn overexpression in P38 phosphorylation is weaker in 4T1 cells than in MDA-MB-231 cells. Despite this difference, we still observed a significant reduction of p-P38 levels by induction of TAK1-dn and a complete abolishment of p-P38 induction with IL1RA in 4T1 cells.

REVIEWERS' COMMENTS:

Reviewer #1 (Remarks to the Author):

The authors have adequately addressed all my concerns. I recommend the revised manuscript is accepted for publication.

Reviewer #2 (Remarks to the Author):

The authors have addressed all my previous concerns. Although they were unable to do the macrophage experiment requested, their explanation was acceptable for why they were unable to do this experiment.